# New Green Approaches in Nanoparticles Synthesis: An Overview

**DOI:** 10.3390/molecules27196472

**Published:** 2022-10-01

**Authors:** Bogdan Andrei Miu, Anca Dinischiotu

**Affiliations:** Department of Biochemistry and Molecular Biology, Faculty of Biology, University of Bucharest, 91-95 Splaiul Independentei, 050095 Bucharest, Romania

**Keywords:** metal nanoparticle, metal oxide nanoparticle, green synthesis, biological method

## Abstract

Nanotechnology is constantly expanding, with nanomaterials being more and more used in common commercial products that define our modern life. Among all types of nanomaterials, nanoparticles (NPs) occupy an important place, considering the great amount that is produced nowadays and the diversity of their applications. Conventional techniques applied to synthesize NPs have some issues that impede them from being appreciated as safe for the environment and health. The alternative to these might be the use of living organisms or biological extracts that can be involved in the green approach synthesis of NPs, a process that is free of harmful chemicals, cost-effective and a low energy consumer. Several factors, including biological reducing agent concentration, initial precursor salt concentration, agitation, reaction time, pH, temperature and light, can influence the characteristics of biologically synthesized NPs. The interdependence between these reaction parameters was not explored, being the main impediment in the implementation of the biological method on an industrial scale. Our aim is to present a brief review that focuses on the current knowledge regarding how the aforementioned factors can control the size and shape of green-synthesized NPs. We also provide an overview of the biomolecules that were found to be suitable for NP synthesis. This work is meant to be a support for researchers who intend to develop new green approaches for the synthesis of NPs.

## 1. Introduction

The global market for nanomaterials has been on a growing trend in the past few years, and economic analysts expect this growth to accelerate in the future. Among all types of nanomaterials, NPs are the most widely produced [1,2]. NPs are included in several products, ranging from nanocomposites in the aerospace and automotive industries to daily products, such as food packaging and additives, sporting goods and textiles, cosmetics, or electronics and IT products. Considering the current progress in research, in the near future, NPs for biomedical [3,4,5] and agricultural [6,7] purposes might become widespread in the global market. Not least, developing countries might be interested in NPs, due to the benefits they bring to the economy [8].

The major reason for which the potential of nanotechnology could be limited is related to the concerns regarding the safety of NPs when they are in contact with human and animal organisms [9]. Despite the evidence that NPs might be harmful, a lot of products incorporating them are already used and, therefore, NPs are produced on an industrial scale. The conventional synthesis techniques usually involve physical and chemical methods, which are disadvantageous in terms of energy consumption and the use and release of pollutant chemicals [10]. 

Biological methods implemented for the synthesis of NPs could be a feasible and more sustainable alternative for the future. Their benefits consist of eco-friendliness, cost-effectiveness and low-energy consumption, in comparison with the conventional methods [10,11]. Green synthesis of NPs uses the reducing potential of the compounds within living organisms. Plants and microorganisms are cosmopolitan, accessible and safe-handling resources. Moreover, the involvement of biomass waste in the green synthesis process can facilitate the transition of NPs production to the principles of a circular economy.

Bio-compounds coat the surface of NPs during green synthesis [12]. Therefore, this additional layer contributes to the enhanced biological properties of green NPs compared to the ones produced using chemical reduction. Moreover, the involvement of bio-compounds prevents the contamination of green NPs with toxic by-products. These can be hazardous compounds that attach to the NPs during the physical and chemical synthesis [13]. However, one of the reasons for which bio-assisted synthesis has remained, so far, in the research stage is that the process involves a set of interrelated parameters (the biological reducing agent composition and concentration, initial precursor salt concentration, agitation speed, reaction time, pH, temperature and light). These impact the characteristics of the formulated NPs. Therefore, the optimization of a biological method is a complex process that needs considerable time and resource. Also, the organisms’ involvement makes the protocols hard to standardize and reproduce, as the internal environment of living beings differs from one species to another and between individuals from the same species. 

The different review articles have focused on the green synthesis of NPs, considering that research interest in this subject has continuously increased in the past ten years [14]. However, some of the published reviews have not referred to the factors affecting the biological synthesis of NPs [15,16,17], while others briefly discussed them [13,18]. Also, the more exhaustive analyses of the green reaction parameters were limited to a single type of NP, such as silver [19], copper [20] or zinc oxide [21,22]. There are no recent in-depth investigations regarding the general conditions that guide the green synthesis of NPs. Therefore, our study was conducted to provide a better understanding of the processes that drive the modulation of NPs’ characteristics during green synthesis. We highlighted the remarkable diversity of biological agents that are involved in nanobiotechnology. Our main aim was to discuss the recently achieved knowledge regarding the factors that influence the biological synthesis of metal and metal oxide NPs. Furthermore, this work could contribute to the drawing-up of guidelines for the development of novel approaches dealing with the synthesis of green NPs. 

## 2. Mechanism of Green Synthesis

There are three ways in which living organisms can be involved in NP synthesis: intracellular (endogenous) synthesis, extracellular (exogenous) synthesis and the use of specific biochemicals. 

The endogenous NPs’ biosynthesis is based on the ability of certain organisms to extract metals from the growing medium and hyperaccumulate them. These microorganisms and plants have been used in biomining activities to extract economically important metals from lands where conventional mining would not have been effective [23,24]. The in-depth research on the bioaccumulation mechanism has shown that plants retain metals in the form of nanometric particles [25]. The synthesis occurs within the cell cytosol, due to its high reductive capacity. Cellular enzymes and other biomolecules take part in the process. For example, Dahoumane et al. set up an innovative approach using living cells of the *Chlamydomonas reinhardtii* microalga to obtain silver–gold alloy NPs. Their study showed that adding silver and gold salts to the algal culture would cause cell sedimentation, due to the internalization of the metallic ions in the cytoplasm. Thus, the reduction process in this case occurs intracellularly; the formed NPs are then released into the extracellular matrix. The last event is very important for the stability of silver–gold NPs, as the extracellular matrix is rich in polysaccharides that act as capping agents [26]. 

Phytomined metals are conventionally extracted by biomass combustion, followed by sintering or smelting. However, these methods are inappropriate for applications that need to maintain the nanometer size of the metallic NPs, because of the extreme temperature and pressure used in these cases [27,28]. In order to maintain the NPs’ structure and properties, methods using mild conditions have been developed. For example, Abdallah et al. developed a method involving different steps of filtration and centrifugation. After the plant material was ground and mixed with 1% SDS, the resulting mixture was filtrated using a sieve. Following decantation, a centrifugation step was performed and the obtained pellet was subjected to lyophilization. Finally, ultracentrifugation in sucrose density gradients was performed in order to separate the NPs’ fraction from the plant debris [25]. Moreover, Marshall et al. proposed enzymatic digestion for the concentration of gold NPs intracellularly accumulated into *Brassica juncea*. They used 1-β-endoglucanase from *Trichoderma viridae* to solubilize the plant biomass for obtaining NPs at a concentration that is suitable for catalysis. However, only 55–60% of the biomass was converted into a soluble form. Even though the NPs were not purified by this enzymatic treatment, their size was maintained [29]. 

It is worth mentioning that the separation of NPs from cells is not always necessary when an endogenous synthesis is performed for catalytic purposes [30,31].

The intracellular NP synthesis imposes some limitations which make it unsuitable for implementation at an industrial scale. The NPs’ morphology cannot be controlled and there are also difficulties regarding the efficiency of NP extraction, isolation and purification [32]. In order to avoid these impediments, the researchers have focused on the development of what we know as in vitro approaches. The mechanism of in vitro NP green synthesis is broadly based on the internal biomolecules from plants and microorganisms able to act as reducing and capping agents. In vitro approaches can involve extracts containing the total biocomposition of the organism used. In vitro synthesis of NPs was also performed by using a single biomolecule, such as quercetin [33], resveratrol [34] or curcumin [35]. 

Living plants can release biomolecules into the environment in response to metal stress. The exogenous synthesis of NPs uses the ability of secondary metabolites released by plant roots to chelate metallic ions. These are secreted to reduce the toxicity of metallic ions, converting them into less toxic nanoscale particles. For example, Shabnam et al. showed that gold NPs were produced when parts of germinated seeds and roots of seedlings of *Vigna unguiculata* were immersed in different concentrations of chloroauric acid. The secretion of phenolic compounds during seed germination and the early development of seedlings might explain the occurrence of the NPs [36]. Similarly, silver NPs have been produced when peanut (*Arachis hypogaea*) seedlings’ roots were exposed to 1 mM silver nitrate [37]. Among all the aforementioned approaches, whole composition extract-based synthesis is used on a large scale. The preparation of extracts generally follows the same steps (Figure 1A). Referring to plant extracts, the protocol usually involves the collection of the selected plant part (1), followed by washing (2) and drying (3). After that, the plant parts are ground (4) and mixed with a solvent at a relatively high temperature (5). The last step consists of filtration of the obtained solvent extract (6) [38,39]. 

Referring to microorganisms, they can be grown in a liquid medium and centrifuged or filtered in order to separate the bio-compounds released into the external environment [40,41,42].

The typical green synthesis reaction simply consists of the mixture between the biological extract and a metal salt solution, which represents the precursor of NPs (Figure 1B). Biomolecules, such as polyphenols (hydroxybenzoic and hydroxycinnamic acids, flavonoids, stilbenes and lignanes) and terpenoids, can transfer electrons to metallic ions, leading to their reduction [33,43]. As a result, the reduced ions start to form ordered arrangements resembling a crystalline structure named nucleus. The nuclei provide support for the reduced ions, which will continue to settle on their surface, leading to the enlargement of the particles [44]. The growth is stopped by biomolecules that act as capping agents, because they attach to the surface of particles, stabilizing their size [45].

The mechanism involving the bio-reduction of ions is generally valid for the formation of metallic NPs. However, this cannot explain the formation of metallic oxide NPs. After bio-reduction, metallic ions such as Zn^2+^, Cu^2+^ or Ni^2+^ could acquire a zero-valent state. Therefore, their conversion to oxides can only be explained by an additional step. Several of the papers have attempted to explain the origin of the oxygen atoms from the metallic oxides produced through green synthesis. Osuntokun et al. proposed that the synthesis of zinc oxide, mediated by *Brassica oleracea* extract, involved two steps. In the first one, chelation of Zn^2+^ cation to polyphenols or flavonoids occurred by the bonding of hydroxyl groups from biomolecules, leading to the generation of zinc hydroxide. In the second step, one molecule of water is released by drying at 70 °C, leading to the formation of zinc oxide [46]. The generation of an intermediate in the green synthesis of metallic oxide NPs was supported also by Velsankar et al., who hypothesized that diethyl phthalate and 1,2-benzenedicarboxylic acid from a *Capsicum frutescens* leaf can interact with the Cu^2+^ cations from copper nitrate to form copper hydroxide. Similar to the mechanism described above, probably the thermal treatment led to the release of one molecule of water and the copper hydroxide being converted to copper oxide [47]. Alternatively, Singh et al. proposed that bio-reduction can occur during the biological synthesis of zinc oxide NPs. They considered that, after Zn^2+^ ions have been reduced to the zero-valent state by phytochemicals from *Eclipta alba*, the Zn^0^ reacted with the molecular oxygen dissolved in the reaction mixture, leading to the formation of zinc oxide [48]. As nucleation and particle growth kinetics are controllable, the formation of particles that fit into the nanoscale can be achieved. Probably the biological extract influences, to the greatest extent, the green synthesis of NPs as its composition and biomolecules’ concentration have a direct impact on the first and last steps of the reaction, i.e., the ion reduction and capping, respectively. Other factors such as the pH value of the mixture can change the electrical charge of the biomolecules [49], while the temperature applied impacts their kinetic energy [50].

## 3. Most Common Techniques Used for Characterization of Biologically Synthesized Nanoparticles

Usually, the reduction of metallic ions determines a color change of the reaction mixture [51,52], thus visual observation can confirm that green synthesis has occurred. A more accurate analysis of NPs can be performed through UV-Vis spectroscopy, which is one of the facile and indispensable techniques for most studies dealing with green nano-chemistry. UV-Vis spectroscopy is mainly used for the primary characterization of biologically synthesized NPs, confirming their formation [53]. The optical extinction spectrum of colloidal solutions of green NPs exhibits an absorption peak that fits into a specific range, depending on the particle composition (Table 1). 

Besides UV-Vis spectroscopy, a series of other techniques is involved in the characterization of novel green NPs. Electron microscopy is widely used to directly observe the shape and size of NPs, while dynamic light scattering (DLS) can estimate their hydrodynamic diameter. Also, the features regarding the crystalline structure of the particles are analyzed by X-ray diffraction. Fourier-transform infrared spectroscopy (FTIR) is another commonly utilized technique, providing information about the functional groups that are present in NPs after green synthesis. The principle of FTIR consists of exposing the sample to infrared radiation, which is then absorbed by the functional groups within its structure. As a result, different types of vibrations, i.e., bending, deformation or stretching, occur. The FTIR spectrum is calculated through a mathematical algorithm based on the initial interferogram. The spectrum displays the absorbance depending on wavenumbers, which are inversely proportional to wavelength. Based on the value of the wavenumber, the functional groups are identified. For example, the FTIR spectra of ZnO NPs, synthesized using *Deverra tortuosa* extract, showed a wide peak at 3434 cm^−1^, suggesting the presence of hydroxyl groups within the structure. Other peaks corresponded to the amines functional groups specific for proteins [92]. 

High-performance liquid chromatography (HPLC) is a useful technique when the mechanism of the formation of green NPs is investigated, especially when plant extracts are involved. The HPLC provides information regarding the composition of the biological agent used in the green synthesis. Different studies have suggested that proteins present in the cell-free supernatants obtained from microbial cultures can act as coating agents [93,94,95]. The proteins implicated in the functionalization of NPs can be identified and characterized by electrophoresis, immunoblotting or sequencing. The knowledge of the sequence of proteins is useful for the understanding of the capping mechanism.

Biomolecules that act as capping agents are important, as they remain attached to the surface of particles, conferring them stability over time. A key measure of the stability of a NP suspension is the zeta potential, which quantifies the electrical charge on the surface of NPs. Generally, suspensions of NPs that have a zeta potential value under –25 and over +25 mV are stable, due to the electrostatic repulsion between strongly charged surfaces. The repulsion between lower-charged NPs is weak, therefore the van der Waals attraction forces can lead to particle aggregation [53].

## 4. Organisms Involved in the Biological Synthesis of Nanoparticles

### 4.1. Bacteria

A vast variety of bacterial species has been used in the synthesis of metallic and/or metal oxide NPs. Several papers have presented strains of different genera with most promising results, such as: *Aeromonas*, *Bacillus*, *Desulfovibrio*, *Enterobacter*, *Escherichia*, *Klebsiella*, *Lactobacillus*, *Pseudomonas*, *Rhodobacter*, *Rhodopseudomonas*, *Shewanella*, *Ureibacillus* or *Weissella* [96]. On this list, other strains of genera isolated from marine environments have been added, including *Alcaligenes*, *Alteromonas*, *Ochrobactrum*, and *Stenotrophomonas* [97]. Different species of *Bacillus,* including *B. subtilis*, *B. pumilus*, *B. persicus* and *B. licheniformis,* have been used extensively to produce silver or gold NPs [41,42], while *B. amyloliquefaciens* has been used for the synthesis of cadmium sulfide NPs [98]. 

Studies are still in progress, thus it is not known yet how many types of metallic ions a bacterial cell can reduce. So far, some bacterial strains have been involved in the production of only one or a few types of NP, while others can mediate the formation of many more types. It is the case of *Pseudomonas aeruginosa*, which is able to synthesize extracellularly seven types of NP, including silver, palladium, iron, nickel, platinum, rhodium and ruthenium. The same microorganism also has the ability to produce large cobalt and lithium particles involving intracellular mechanisms [99]. 

Most of the bacterial strains involved in NP synthesis are terrestrial organisms. Reference strains, as well as isolates from specific environments (e.g., mines) where bacteria may enhance their native ability to bioleach and bioaccumulate metals, have been used. However, the recent studies have reported the use of marine bacterial cultures in the synthesis of silver, gold, copper and cadmium sulfide NPs [97]. For example, the extracellular polymeric substances (EPS) extracted from a marine strain (JP-11) of *P. aeruginosa* were successfully used to produce cadmium sulfide NPs. The sulfhydryl functional groups of the EPS play a major role in the mechanism of NP formation, as they represent binding sites for metallic ions [100]. 

The manipulation of genetic material is easier when it comes from bacterial organisms, rather than other living beings. Bacterial genes have already been used in transgenic approaches aiming to obtain eukaryotic cell cultures able to generate NPs. Therefore, the *mms6* gene, which contributes to the formation of magnetosome in *Magnetospirillum magneticum* bacteria, was inserted in human mesenchymal stem cells, and the intracellular formation of iron NPs with magnetic properties was revealed. NPs with sizes between 10 and 500 nm were accumulated in vacuoles [101].

### 4.2. Microfungi and Actinomycetes 

In recent years, many fungal strains were studied for their ability to reduce metallic ions. Some of the literature reviews summarized the genera with the most important results, such as: *Aspergillus*, *Cladosporium*, *Colletotrichum*, *Fusarium*, *Penicillium*, *Phoma*, *Trichoderma* and *Trichothecium* [102]. Other studies have focused on biosynthesis, using ascomycetes of the *Brevibacterium*, *Corynebacterium*, *Kocuria* and *Neurospora* genera [96]. 

The group of fungi that produce NPs is completed by some actinomycetes, also known as mycelial bacteria, as they share some characteristics with prokaryotes. The NPs synthesized by actinomycete-mediated approaches appear to be stable, with good polydisperse distribution and high bactericidal activity. However, the potential of actinomycetes to be involved in NP production has not been explored at a high capacity. These microorganisms have been used extensively in the biosynthesis of silver and gold NPs, while a few of them have been involved in the formation of zinc and some bimetallic NPs. The most studied actinomycetes in nanobiotechnology belong to the *Nocardia*, *Nocardiopsis*, *Rhodococcus*, *Streptomyces*, *Thermoactinomyces* and *Thermomonospora* genera [103]. 

### 4.3. Yeasts

Yeast-mediated approaches have been used extensively to synthesize cadmium sulfide or lead sulfide NPs. However, the ability of yeasts to synthesize gold, titanium dioxide, antimony trioxide (Sb_2_O_3_) or other NPs has also been proved [104]. The production of silver NPs has been performed using different strains of *Candida*, including *C. albicans*, *C. glabrata*, *C. lusitaniae* and *C. utilis* [97]. The most important studies performing yeast-mediated synthesis of different NPs were cited by Narayanan and Sakthivel [102]. These works used yeast organisms such as *Pichia jadinii*, *Saccharomyces cerevisiae*, *Schizosaccharomyces pombe* or *Yarrowia lipolytica* [102]. The recent papers cited by Gahlawat and Choudhury [97] complete this list. These include organisms belonging to the *Cryptococcus*, *Magnusiomyces*, *Phaffia*, *Rhodosporidium* and *Rhodotorula* genera [97].

### 4.4. Microalgae and Cyanobacteria

Microalgae can be used for various biotechnological applications. They were first studied for the removal of nutrients and organic carbon, as well as heavy metals, from wastewater. Also, their potential to be used in the manufacture of commercial products or biofuels has been the subject of research studies over time [105,106]. The advantages of algal cultures have recently granted them the status of candidates for the greener synthesis of NPs. Not only their high tolerance to metallic ions [107], but also their high growth rate and efficiency in producing different biomolecules [108], has led to greater attention paid to microalgae in the nanobiotechnological research. 

*Chlorella vulgaris* seems to be the most studied microalga for its ability to reduce metal ions. So far, this has been involved in the green synthesis of many types of NPs [97,109]. Other *Chlorella* species, namely *C. pyrenoidosa* and *C. kessleri*, have also proved to be effective as nanofactories. They have been involved in the synthesis of silver and/or copper NPs [110,111,112]. Also, *Botryococcus* sp., *Chlamydomonas* sp., *Coelastrum* sp., *Scenedesmus* sp., *Neochloris oleoabundans*, *Galdieria* sp., *Dunaliella tertiolecta* or *Tetraselmis suecica* [106,110] represent other examples of microalgae used recently to produce metallic NPs. Diatoms represent a group of unicellular algae distinguishable by micro- and nanopatterned siliceous cell walls. These structures are formed by the deposition of silica, biogenerated from orthosilicic acid taken-up from the environment [113]. Different species of diatoms, such as *Stephanopyxis turris* [114], *Amphora copulate* [115], *Navicula atomus* [116] and *Diadesmis gallica* [117], have been involved in the production of gold NPs, while *Navicula* sp. [118], *Chaetoceros* sp., *Skeletonema* sp. and *Thalassiosira* sp. [119] have been used for the generation of silver nanostructures. Due to their unique 3D structure, diatoms can be used as scaffolds for the anchoring of metallic NPs, hence the occurrence of biosilica-based nanocomposites for drug delivery [120], biosensing [121] and catalysis [117] purposes. 

Cyanobacteria represent a large taxonomic group containing photosynthetic organisms. As with other microorganisms, these green bacteria were found to be suitable for the biological synthesis of nanoscale particles. Even though their nanotechnological potential has not been well explored, some studies focused on cyanobacteria-mediated NP synthesis have emerged recently. The ability to reduce metallic ions was found in cyanobacterial species belonging to the *Anabaena*, *Cylindrospermopsis*, *Limnothrix*, *Lyngbya*, *Synechococcus*, *Synechocystis* and *Arthrospira* genera [110,122].

### 4.5. Plants

Several living plants have been explored extensively for their ability to accumulate heavy metals, thus remedying contaminated environments. The researchers in nanobiotechnology have paid more attention to the plants with phytomining potential, after it was discovered that they store metals in the form of nanoscale particles inside their tissues. Mohammadinejad et al. recently listed the most important plants involved in the endogenous mediated synthesis of NPs, including *Sesbania drummondii*, *Ipomoea lacunosa*, *Festuca rubra* and *Arabidopsis thaliana*. The most studied plant species for the production of NPs inside its tissues is *Medicago sativa,* followed by *Brassica juncea*. These species have been involved, so far, in the synthesis of silver, gold, copper and platinum NPs [123]. Even though the list of plants studied for endogenous NP synthesis is not so wide, there are many plants with phytomining potential that have not yet been involved in nanotechnological approaches (e.g., *Acanthopanax sciadophylloides*, *Maytenus founieri* and *Clethra barbinervis*) [123,124]. 

Regarding the synthesis of NPs using plant extracts, the research teams have focused on plants belonging to the angiosperm taxonomic group. The diversity of plant extracts that have been involved so far in NP synthesis is remarkably high. For example, extracts from more than 30 different plant species have been used only for the synthesis of titanium dioxide NPs [125]. It seems that medicinal plants represent a priority, as their biomolecules already possess healing properties that may be transferred to NPs through the capping process. In a recent review, Agarwal and Gayathri [126] identified at least 25 studies exploring the potential of medicinal plants to produce different types of NPs, most of them composed of silver or zinc oxide. The most explored property was their antibacterial activity, while a few studies focused on the anticancer, antileishmanial, antioxidant or anticoagulant potential of green NPs. Examples of medicinal plants used as nanofactories are: *Prunella vulgaris*, *Suaeda maritima*, *Bauhinia acuminata*, *Taraxacum laevigatum*, *Carum copticum* and many others [126]. 

Besides the species with medical importance, some plants used as food have received attention in nanotechnological research. For example, different types of fruit trees, including *Malus domestica* (apple tree), *Citrus sinensis* (orange tree), *Juglans regia* (walnut tree) and *Prunus persica* (peach tree), have been involved in the biological synthesis of silver, titanium dioxide and zinc oxide, respectively, and iron oxide NPs [127,128,129,130]. The synthesis of iron oxide NPs has also been performed using beet (*Beta vulgaris*) and pumpkin (*Cucurbita moschata*) extracts [131], while spice plants such as rosemary (*Rosmarinus officinalis*) and ginger (*Zingiber officinale*), respectively, have been used to produce iron- and copper-containing bimetallic NPs [132,133]. Not least, several angiosperms with no specific importance to humans have proved to be effective as mediators of nanoscale particle synthesis (e.g., *Eichhornia crassipes*, *Linaria maroccana*) [134,135]. 

Besides the angiosperms, the Kingdom Plantae also includes gymnosperms, pteridophytes (ferns), bryophytes (mosses) and macroalgae. All of these groups contain at least one representative that has been explored for its ability to mediate the production of NPs, especially the ones made of silver, gold, platinum or palladium. Das et al. [136] have recently managed to include all these organisms in a broad and comprehensive literature review. They found that the most studied gymnosperms for the green synthesis of NPs are: *Cycas circinalis*, *Ginkgo biloba* (also used for medicinal properties), different species of *Pinus*, *Thuja orientalis* and *Torreya nucifera*. Regarding the pteridophytic plants, there are at least three genera that have been explored so far for nanobiotechnological applications: *Adiantum*, *Pteris* and *Nephrolepis*. The bryophytes are also less explored. To date, some approaches trying to obtain silver NPs using extracts from *Anthoceros* sp., *Riccia* sp. or *Fissidens minutus* have been reported. Compared to the mosses and ferns, the macroalgae are better represented in nanobiotechnological research. The studies dealing with the green synthesis of NPs have used macroalgal organisms classified as: *Colpomenia sinuosa*, *Pterocladia capillacea*, *Jania rubens*, *Ulva fasciata*, *Ulva intestinalis* and different species of *Sargassum* [136].

## 5. Factors Affecting the Biological Synthesis of Nanoparticles

### 5.1. Reducing Agent and Precursor Salt Nature and Concentration

Increasing the concentration of the reducing agent would normally accelerate the growth of particles. Thus, highly concentrated biological extracts would lead to the formation of larger NPs. For example, the study of Kumari et al. [137] showed that increasing the concentration of fungal filtrate of *Trichoderma viride* from 10% to 100% would increase the size of gold NPs by almost six times. The same tendency was indicated by the absorption spectra analyses of silver NPs, when different concentrations of *Ocimum sanctum* [43] or *Plantago major* [138] extracts were used. 

The increased size of NPs might be explained by a secondary reduction process that occurs on the metal nuclei, because of the excess of reducing phytochemicals. Nagar and Devra [139] observed that exceeding a concentration of 20% of *Azadirachta indica* plant extract causes the agglomeration of copper NPs, thus larger structures are formed. However, a low reductant concentration (<5%) does not provide enough biomolecules to start the conversion of ions [139]. Even if a low level of biomolecules could be enough to trigger the reduction step, they would be rapidly depleted by the metallic ions. This would cause an insufficient amount of biochemicals to be involved in the capping process, and so the aggregation of particles and precipitation could occur [140]. On the contrary, an excessive amount of biochemicals can impair the nucleation step [141]. 

The reducing power of biological extracts is linked to their composition and amount of biochemicals. Therefore, the solvent used for obtaining the extract can impact the green synthesis of NPs. A recent study showed that the average size of silver NPs, synthesized using the ethanolic extract of *Acacia cyanophylla*, almost reached 200 nm. On the contrary, NPs obtained in the same conditions, but with the aqueous extract, had an average size of around 87 nm [142]. As ethanol is less polar than water, the biomolecules are expected to be more concentrated and also diverse in the ethanolic extracts compared with the aqueous one. The variety and higher content of molecules in the ethanolic extracts from different plants were also confirmed experimentally [143,144]. 

The synthesis of palladium NPs prepared via a methanolic extract of *Eryngium caeruleum* took place in 60 min, while the ethanolic extract-mediated synthesis needed 100 min to be completed. Also, the formation of the same NPs took a couple of hours when the aqueous extract was used [145]. 

The biomolecules involved in the green synthesis of NPs differ, depending on the organism used for their extraction. However, this is not valid for the precursors, which are simply salts containing the ion of interest. Most studies describe a decrease in the average size of NPs, which is proportional to the increase in the precursor concentration [137,139,146]. When the precursor amount is high, more nuclei were formed and the capping agents acted quickly to stabilize them. However, when the concentration of the precursor was too high, the level of phytochemicals became insufficient to stabilize a large number of nuclei. 

Thus, the larger NPs are formed due to the aggregation of nuclei. Nagar and Devra showed that the increasing of the concentration of copper chloride (CuCl_2_) up to 7.5 mM decreases the NPs size to almost 45 nm. Over the aforementioned concentration value, the size of the copper NPs exceeds 75 nm [139]. Also, in the case of silver NP synthesis, the increase of silver nitrate (AgNO_3_) amount in the reaction mixture leads to the formation of larger NPs, as revealed by spectroscopic analyses [146,147]. 

Similar results were obtained for gold NP synthesis. Using chloroauric acid (HAuCl_4_) mixed with *T. viride* fungal extract, Kumari et al. [137] showed that the growth of particles is promoted by a high precursor concentration. For example, at 30 °C and a concentration of 250 mg/L HAuCl_4_, the average size of the gold NPs was 34 nm, while at 500 mg/L HAuCl_4_, the NPs were 85.2 nm in size. At 50 °C and the same concentrations of gold salt previously mentioned, the mean size of the particles increased from 273.6 nm to 699 nm. In both cases, the average size increased by about 150%, proving that the metal salt concentration works together with other reaction parameters to mediate the size of NPs [137]. 

Different precursor salts can be utilized to obtain a single type of NP (Table 2). Therefore, the properties of NPs produced via green approaches can be influenced simply by the chosen precursor. However, the information regarding the effect of different precursor salts on green NP characteristics is limited. Droepenu et al. revealed that ZnO nanospheres with a diameter of approximately 107 nm have been obtained when zinc acetate and *Anacardium occidentale* leaf extract were mixed. The use of zinc chloride has led to the synthesis of ZnO nanorods that were 167 nm in length and 68 nm in width. However, both samples faced aggregation [148]. The ZnO NPs synthesized from zinc sulphate had a nanorod morphology and an average size of 30 nm when *Justicia adhatoda* leaf extract was used. Contrariwise, the NPs obtained by the same method, but starting from zinc nitrate or zinc acetate, were cubic and 15–20 nm in size. Moreover, their tendency to form agglomerates was observed. Interestingly, the use of different precursors can have an impact also on the biological activity of NPs. The antimicrobial activity of ZnO NPs obtained from zinc nitrate was more pronounced against bacterial strains, while the ones produced using zinc sulphate were more efficient against different strains of *Aspergillus* [149]. The influence of different precursors on ZnO NP morphology was also confirmed by Fakhari et al., using *Laurus nobilis* leaf extract as a reducing agent [150]. Fatima et al. revealed the effect of different precursors on the shape of iron (III) oxide particles using synthetic reagents. Cubic and octahedral particles were formed from ferrous sulphate, solubilized in ethylene glycol and capped with potassium hydroxide. On the contrary, spherical particles occurred when ferric chloride was used instead [151]. 

### 5.2. Agitation Speed

Agitation is important to keep reactants in motion. The chance of metal ions to come into contact with biomolecules is higher in a continuously stirred mixture. Therefore, the relationship between the rate of the green synthesis reaction and the agitation speed should be directly proportional. This role of agitation was confirmed when iron NPs were synthesized using fungal biomass. The UV-Vis analyses revealed that absorption of the agitated mixture was double in comparison with that corresponding to the static one [174]. The fact that agitation speed can increase the reaction rate is also supported by the study of Selvakumar et al., in which a silver NP synthesis was based on the molecules found in the leaves of *Acalypha hispida*. The formation of NPs was faster as the stirring speed increased, reaching 5–7 min at 700 rpm. However, the results pointed out that excessive speed could affect the biomolecules involved in the green process. This could be the reason for which the reaction rate was negatively affected at an agitation speed over 700 rpm [175]. 

Agitation speed may also have an impact on the characteristics of NPs. Chan and Don revealed that stirring the reaction mixture at 100 rpm led to the formation of silver NPs with sizes over 80 nm, while increasing the speed to 250 rpm caused the formation of NPs with a diameter around 15 nm [176]. Some computational statistical optimizations also confirmed that increasing the agitation speed led to a decrease in the size of the NPs [177,178]. 

### 5.3. Reaction Time

The variation in reaction times can be used to control the size and shape of green-synthesized NPs. During the biological synthesis of NPs, three main events occur: (i) the initiation of the ion reduction process, (ii) the nucleation and growing of the NPs and (iii) the complete reduction of ions. There is evidence that the initiation of the reduction process requires between 5 and 15 min [146,179] while, in some cases, it occurs immediately after the precursor salt is mixed with the biological extract [180,181]. A complete reaction can occur in between 45 and 120 min, depending on the reducing agent’s effectiveness [141,146,179]. However, some studies have proved that the green reactions may reach equilibrium after one week [182,183]. For example, Wei et al. [184] reported that silver NPs needed five hours to be completely formed in a mixture of organic residues while, in *Hibiscus cannabinus* leaf extract, they became stable after five days [185]. 

The increase in reaction time is directly proportional to particle size and number of generated nuclei. If the complete reduction time is exceeded, the NPs start to aggregate and form larger structures. For example, after 24 h, Kumari et al. observed that *T. viride* fungal extract-mediated gold NPs had a spherical shape and a size between 7 nm and 24 nm. Increasing the incubation time to 72 h led to the formation of NPs with a size between 20 nm and 400 nm. Moreover, the gold NPs changed their shape, becoming triangular, and also nanoprisms were observed. It was stated that the crystal growth was enhanced by time. The initial nanospheres fused and formed triangular-shaped NPs, which further fused and, as a result, the nanoprisms were formed [137]. Similarly, silver NPs obtained from a 24-h green reaction had a size of around 62 nm, while their diameter exceeded 100 nm after 72 h [142].

### 5.4. Reaction pH

Biomolecules involved in the green synthesis of NPs could have different reducing activity levels based on the pH value of the reaction mixture. Alkaline pH conditions cause the deprotonation and activation of phytochemicals, while lower pH values constrain them to remain mostly protonated, leading to a decreased reducing or capping activity [49,186,187]. The inactivation of phytochemicals in an acidic environment is supported by some studies reporting no NP formation at a very low pH [49,139,188,189]. For example, the formation of silver NPs via biological methods cannot take place at pH 2 [188] or 3 [189]. Also, the green synthesis of copper NPs cannot occur in a reaction mixture with a pH of 4.7. However, the synthesis of green copper NPs is more efficient in a mildly acidic environment (pH 6—6.6), because of the agglomeration that takes place at a higher pH value [139]. On the contrary, Din et al. reported that, at pH values between 2 and 8, the leaf extract of *Calotropis gigantea* was not able to reduce the Ni^2+^ ions provided by the nickel nitrate. The formation of nickel and nickel oxide NPs occurred only at a strong basic pH, with values between 10 and 12 [49].

In general, NPs produced in an alkaline reaction mixture have a smaller size and are more stable in time. The reason might be the capping process that occurs earlier and is more efficient at alkaline pH values, due to a large quantity of activated phytochemicals. Using the *Parachlorella kessleri* microalgae, Velgosová et al. have reported a wide histogram of the size distribution for silver NPs produced at pH 4, with a minimum of 20 nm and a maximum of 60 nm. By comparison, silver NPs had an average diameter of 15 nm and a size range between 10 and 20 nm when their formation took place at pH 10 [188]. The same tendency was observed when silver NPs were synthesized using green tea (*Camellia sinensis*) extract. The average size of NPs formulated at pH 5.8 was around 50 nm. It successively reduced when the pH value increased, the NPs reaching ≈28 nm in diameter [190]. Similar results were obtained in the case of *Amomum* sp. synthesized gold NPs. At pH 3, the obtained NPs were larger (90–100 nm) than those produced at a pH value of around 7 (20–40 nm) [180]. 

On the contrary, there is also evidence that NPs synthesized in strong acidic environments can have smaller diameters than the ones produced at mildly acid pH values. Prabhakar et al. reported the aqueous leaf extract of *Eichhornia crassipes* led to the formation of iron NPs with a size between 20 and 60 nm [134]. Almost the same NP morphology was reported when *R. officinalis* leaf extract was used [132]. However, when a more alkaline leaf extract was used, the analyses showed that larger iron NPs (60–200 nm) were formed. For comparison, larger NPs were obtained using the leaf extract of *Mimosa pudica*, which has an average pH of 6, compared to *E. crassipes* that has an average pH of 4 [134]. The pH value of aqueous plant extracts is mainly dependent on the amount of solubilized organic acids. As *R. officinalis* is rich in rosmarinic and carnosic acids [191], probably its pH value is also situated in the same range with that of *E. crassipes* extract. 

Even if the pH value can be used to adjust particle size, other properties of NPs can be affected. As expected, the NPs produced at an acidic pH are usually less stable because of the inefficient capping process. Velgosová et al. [188] showed that silver NPs produced at an alkaline pH were unchanged three weeks after their formation. By comparison, the size of the NPs formed in acidic solutions continued to increase over time, and some aggregates were also observed [188]. Probably, the pH applied during the green synthesis influences the electric charge on the NPs’ surface, affecting their stability. The experiments performed by Manosalva et al. [192] show that, at a high pH, the zeta potential of green Ag NPs is lower than −30 mV. Normally, the surface charge of these NPs is strongly anionic, giving them high and long-lasting stability. 

The influence of pH on NPs’ shape is not well understood so far. Singh and Srivastava showed that there is no correlation between the pH and bio-inspired gold NPs shape, even though, at an acidic pH, the NPs had a spherical flower shape and, at an alkaline pH, some triangular NPs were observed [180]. Kumari et al. also reported that gold NPs changed their morphology when the pH value increased. While the NPs obtained at pH 5 had a variable shape, they became triangular when the applied pH had a value between 6 and 8, and spherical at pH 9 [137].

### 5.5. Reaction Temperature

There are two stages within green synthesis methods when the temperature applied can impact the physicochemical characteristics of the formed NPs: the reaction itself and the drying of the particles. A reaction temperature increase leads to a higher kinetic energy of the biomolecules involved in the reducing process, thus the metallic ions are consumed faster. Therefore, the high temperature causes the formation of smaller particles; this fact has been confirmed experimentally by a series of studies. There is evidence that the biological synthesis of Ag NPs cannot occur in *Ocimum sanctum* leaf extract at 5 °C [43]. Also, the *Hygrophila spinosa* extract-mediated synthesis of gold NPs was not effective at room temperature [141]. However, gold NPs were obtained at room temperature when other plant extracts were used [193,194]. Absorbance spectra analyses suggested that an increase of temperature from 15 °C to 35 °C could cause a decrease in the silver NP size [43]. Shankar et al. obtained a similar result, analyzing the effect of temperature on the synthesis of silver NPs mediated by *Rhodomyrtus tomentosa* extract. The synthesis performed at 28 °C resulted in the formation of NPs that were 30% larger in diameter than those obtained at 50 °C, according to the DLS analysis. Furthermore, the quantity of recovered NPs from the reaction medium was higher and the capping process was more efficient at the high temperature [195]. However, extreme temperatures could affect the properties of NPs, probably due to the degradation of the phytochemicals involved in the synthesis process. Using the extract of *Hygrophila spinosa* to synthesize gold NPs, Satpathy et al. found that the optimum temperature of the reduction process is around 80 °C. UV-Vis spectra showed that, at temperatures above 80 °C, the NPs would form aggregates, suggesting an enhanced surface activity of the formed nuclei and an inefficient capping process. Also, it seemed that the formation of gold NPs was not completed at temperatures below 80 °C [141].

The same parameters are valid for biologically synthesized copper NPs. The conversion of Cu^2+^ ions to copper NPs is significantly higher when the temperature increases in the range between 60 °C and 85 °C. At temperatures above 85 °C, the copper nuclei would turn into agglomerates because of the growth rate of NPs, which is surpassed by the nucleation rate [139]. 

The temperature applied for the drying of metallic oxide NPs can also have an impact on the particle morphology, and it is crucial for their crystalline structure. Bala et al. showed that the drying temperature of 30 °C is not high enough for a complete formation of the crystalline structure of ZnO; higher temperatures (60 °C or 100 °C) are necessary for a complete crystalline structure [196]. The same condition is valid for TiO_2_ NPs, which exhibit different crystalline phases during calcination. Usually, the transformation of the anatase phase into rutile begins at temperatures between 600 and 700 °C [197]. However, it is considered that the transition to rutile needs higher temperatures if the TiO_2_ NPs are synthesized using *Peltophorum pterocarpum* extract [198]. The crystallinity of green nickel oxide NPs was also increased at higher temperatures, between 300 °C and 500 °C [199]. 

The drying temperature can also have an impact on the final size of NPs. Dried at 60 °C, ZnO NPs presented a spherical shape and a size between 16 and 60 nm. If the drying process occurred at 100 °C, the particles exceeded the size of 100 nm [196]. This fact was also observed by Maensiri et al., who developed a simple laboratory method which used *Aloe vera* leaves extract and indium acetylacetonate as a precursor for the synthesis of indium oxide NPs. The control of the NPs’ size by calcination at different temperatures in the post-synthesis step was achieved. Therefore, three sets of indium oxide NPs were obtained: (1) within the size range 5–10 nm calcinated at 400 °C, (2) within the size range 10–25 nm calcinated at 500 °C, and (3) within the size range 30–50 nm calcinated at 600 °C [200]. The dimensions of the NPs seemed to increase in a temperature-dependent manner. 

Scanning electron microscopy (SEM) images of cobalt oxide NPs synthesized using *Piper nigrum* extract reveal that excessive temperatures cause cluster formation. Therefore, the size of cobalt oxide NPs successively increased from around 22 nm (at 100 °C calcination) to ≈ 77 nm (at 900 °C calcination) [201].

### 5.6. Light Exposure

There is evidence that the biological synthesis of some photoactive NPs, such as silver or gold, is a light-dependent reaction. The necessity of light exposure in green synthesis was demonstrated by studies reporting no NPs formation in dark conditions. Srikar et al. showed that the formation of silver NPs did not occur in dark conditions when *Prunus amygdalus* fruit extract was used as a reducing agent [202]. This result was confirmed by the study of Kumar et al., who used the leaf extract of *Erigeron bonariensis* [147]. 

Light radiation can have an influence on the green synthesis of NPs due to its intensity and wavelength. Light intensity is proportional to the NPs’ rate of formation. Placing the reaction tubes in direct sunlight will speed up the production of silver NPs 60 times compared to the case when the reaction takes place in diffuse light [202]. A more detailed study has confirmed that the increase in light intensity leads to the maximum production also for gold NPs. Using *Shewanella oneidensis* bacteria as the reducing agent source, the amount of gold NPs increased fast in bright light (>10,000 lux) conditions. Furthermore, the concentration of gold NPs in the test tube exposed to the highest light intensity was almost double compared to that maintained at low light after 3 h of exposure. The reaction did not start in dark conditions, showing that the presence of light is essential for gold NPs formation [203]. 

There is less knowledge about the light wavelength impact on the synthesis efficiency of silver or gold NPs. Srikar et al. found that violet light (380–450 nm) contributes the most to accelerating the biosynthesis of silver NPs. However, better results were reported when all the visible spectrum was applied [202]. Blue light (425–490 nm) stimulated the largest number of functional groups on the surface of *S. oneidensis*, enhancing the formation of gold NPs [203]. Also, blue light significantly reduced the time needed for the complete formation of silver NPs within the *Prunus serotina* fruit extract [204]. According to these results, the biosynthesis of specific NPs seems to be more efficient when the reaction occurs at low wavelength visible light. 

Recently it was reported that the biofabrication of silver and gold NPs can take place also under low UV light [205]. Moreover, Lomeli-Rosales et al. proved that microwaves are an appropriate source of radiation for the green synthesis of gold NPs [206]. 

Regarding exposure time, the reaction starts within the first 5 min in a high-intensity light source, and there is evidence that the formation of NPs cannot be completed in less than half an hour. The increase in exposure time leads to the formation of more particles, but a longer time would cause the aggregation of the NPs [147].

Some studies have tried to explain the mechanism by which light irradiation promotes the green synthesis of some NPs. It has been shown that only the conversion of metallic ions to NPs can be light-dependent. The adsorption of ions to phytochemicals or the capping of NPs are processes that can occur even under dark conditions [207]. Most probably, the light-induced synthesis mechanism is based on photon flux, which may activate the chemical groups of biomolecules involved in electron transfer [203]. NPs containing silver or gold possess photocatalytic activity. After the formation of the first NPs, these may be photoexcited, acting as catalysts for ion reduction. This hypothesis may explain the enhancement of NPs formation when light is applied [207].

### 5.7. Biomolecules Involved in Green Synthesis

Even though the reaction between biological extracts and ionic compounds is not fully understood, some studies provide insight into the biomolecules that drive the green synthesis of NPs. Arsiya et al. revealed that chemicals containing amide and polyol functional groups from the unicellular organism *C. vulgaris* were involved in the reduction of Pd^2+^ ions, leading to spherical NPs with an average size of 15 nm [109]. FTIR measurements showed that biomolecules containing carbonyl and hydroxyl groups from *Solanum nigrum* leaf extract were involved in the green synthesis of palladium NPs. Kaempferol, luteolin and gentisic acid, which are present in *S. nigrum*, were possibly responsible for the reduction of Pd^2+^ ions. The formed palladium NPs had a spherical shape and a size between 3 and 35.7 nm, but they were not effectively capped by the respective phytochemicals [152]. The approach of Ismail led to the production of copper NPs with a size between 7 and 10 nm, and, most probably, the hydroxyl and carbonyl groups from the *Rhus coriaria* fruit were involved in the capping process. As revealed by the FTIR analysis, the structure of these functional groups is especially found in a series of glucosides of antocyanidins (cyanidin, peonidin, pelargonidin) and flavonols (myricetin, quercetin) [155]. 

The study of Ghidan et al. showed that copper oxide NPs were capped mainly with proteins linked through the hydroxyl and carbonyl groups when the aqueous extract of *Punica granatum* fruit peel was used [208]. Moreover, Singh et al. have also highlighted that the capping process of gold NPs synthesized using the aqueous extract of *Dunaliella salina* was most probably managed by proteins [209]. 

Furthermore, different polyphenols were involved in the reduction of Au^3+^ during a plant-mediated synthesis. Ascorbic, gallic and caffeic acids are polyphenolic compounds normally found in the *Sansevieria roxburghiana* leaf extract. HPLC analyses detected the amount of ascorbic and gallic acids significantly decreased after the reduction of the gold ions, while the caffeic acid was completely consumed. The morphology of the produced gold NPs was assessed by transmission electron microscopy images that showed mostly spherical particles with an average diameter of 17.48 nm [210]. It is considered that polyphenols of pumpkin leaves (*C. moschata*) and beet stalks (*B. vulgaris*) form a complexation system through which the Fe^2+/3+^ ions are bound. This system stabilizes the iron nuclei in the oxide form, arranging around them and forming a protective layer [131].

Even though FTIR measurements have highlighted the role of small molecules such as phenols and proteins in the synthesis process, it seems that NADPH-dependent reductases from fungi are crucial in the reduction of AgNO_3_. Experiments have shown that the dialyzed fungal filtrate of *C. cladosporioides* containing cellular proteins cannot reduce silver salt, and the same situation happens in the case of NADPH alone. But when both solutions were applied, the formation of silver NPs took place [211]. Hulikere et al. demonstrated the involvement of NADPH-dependent enzymes also in the case of Au^3+^ ions’ reduction [212]. Regarding the stabilization of silver NPs, more important are phenols, tannins and flavonoids. These biomolecules can bind metals, with their presence on the surface of the silver NPs being suggested by the FTIR analysis performed by Jini and Sharmila [213]. Silver NPs have been generated using an aqueous onion (*Allium cepa*) extract, with the whole process being carried out at room temperature. A SEM analysis has revealed highly stable and uniformly distributed silver NPs with a diameter size between 49 and 73 nm [213]. 

In general, whole organism extracts are preferred among the majority of the studies, probably due to their simple way of implementation and low cost. However, variation in the composition of biological extracts could be an impediment to the standardization of the green synthesis of NPs and the reproducibility of the developed approaches. Moreover, the utilization of whole organism extracts does not allow the analysis of the impact of different molecules on the green synthesis of NPs. Some one-molecule green approaches that have recently emerged might provide an insight into the suitability of biochemicals for green synthesis of NPs (Table 3). 

Among the phytochemicals used in the green synthesis of NPs, eugenol is one of the terpenoids that has proved to be helpful in the production of NPs. Tekin et al. reported the obtaining of silver NPs with a cubic shape and dimensions between 20 and 30 nm [224]. Also, lycopene, a major terpenoid present in tomatoes, can act as a ligand for titanium ions when combined with titanium tetrabutoxide. Baskar and Nallathambi proposed that the mechanism of green titanium dioxide NPs formation involves the cleavage of a double bond C = C within the structure of the lycopene. Afterwards, one titanium ion interacts with one of the C atoms, while O interacts with another one, thus a titanium dioxide-lycopene complex is generated [225]. 

Moreover, it seems that other phytochemicals, such as organic acids, are involved in the reduction of metallic ions into NPs, due to the release of hydrogen when the enol group undergoes transformation to the keto group. Thereby, organic acids act as electron donors for the reduction of ions. This is the case of gallic acid within some hydroxyl groups passing through keto–enol tautomerization, while other hydroxyls are involved in the formation of hydrogen bonds, stabilizing the structure of the NPs. The keto–enol tautomerization occurs through two electrons oxidation at a normal pH [217]. The same mechanism is valid for the tannic acid-mediated synthesis of NPs [219]. 

Besides the organic acids, some studies have proved that hydroxyl groups are also involved in the ion reduction process driven by flavonoids and other polyphenols. Probably, the reactivity of biomolecules is dependent on the number of hydroxyls within their structure. By comparing the FTIR measurements, Das et al. confirmed that the hydroxyl groups within epigallocatechin-3-gallate, a major polyphenol in green tea, were the ones involved in the reduction of Ag^+^ [215]. The FTIR analyses also revealed that phenolic hydroxyl groups of silymarin are involved in the stabilization of gold NPs, being attached to their surface [230], probably by electrostatic attraction.

Curcumin, originating from turmeric, is another polyphenolic compound that has proved to be effective in synthesizing different types of NPs. Iron oxide NPs synthesized via curcumin had a relatively high stability, considering their zeta potential of –35 ± 2.5 mV, while their size was less than 20 nm [239]. There is evidence that curcumin is efficient in the synthesis of copper oxide NPs when its concentration, at most, equals the precursor’s one [240]. 

The impact of the concentration of biomolecules on the characteristics of the formed NPs should follow the general rule presented above (see Section 5.1.). As the concentration of gallic acid increased from 0.1 to 5 mM, the silver NPs increased from 35 to 79 nm and the same tendency was reported in the case of the gold NPs [217]. However, by studying the efficiency of different extracts in synthesizing cerium oxide NPs, Iqbal et al. found that the particle size was smaller when a higher level of quercetin in plants was noticed [241]. However, this result could be explained by the interference of the reducing power of the other biomolecules present in the extracts (e.g., eugenol, furfural and caffeine).

## 6. Conclusions and Future Prospects

Environmental issues force researchers to develop new eco-friendly strategies to produce the goods we need in human society. The future of NPs in industry, medicine and agriculture is expected to be extraordinary, but also uncertain if their environmental issues are taken into consideration. The disadvantages and unsustainability of conventional techniques for the synthesis of NPs have not attracted the public’s attention so far. However, the constant growth of the use of NPs in day-to-day goods require us to think about better alternatives to these. At the moment, the research has demonstrated that bringing living organisms into the nanotechnology area is possible and useful. Progress is constantly being made and researchers have identified new suitable organisms for NP synthesis, taking into account the extremely large diversity of our biosphere. Further efforts to decipher how to control NP characteristics must be made, because this is how green NPs could become attractive for industry, avoiding the negative impact of the growing nanotechnological processes on the environment. 

We consider that the future research should focus less on discovering new organisms suitable for the green synthesis of NPs, and more on the improvement of the reaction itself. In order to achieve this objective, the understanding of the interrelation between the factors is required. Broadly, it is known that an alkaline pH or raising the temperature leads to smaller NPs, but there are also in-depth mechanisms that need to be explained. In the ideal scenario, we would need to know how to manipulate all the parameters involved in the biological synthesis process to adapt the characteristics of the NPs. Therefore, we would be able to design NPs with a suitable activity, according to the application in which they are meant to be used. We assume that one-molecule approaches will be more used due to some advantages. The utilization of a single biomolecule into the synthesis of NPs prevents the variability of the natural extracts, encouraging the standardization of biological methods and their reproducibility. Moreover, these approaches are expected to contribute significantly to the understanding of the function of specific molecules in the mechanism of synthesis. However, one-molecule approaches eliminate the possible synergism between different biomolecules of living organisms. This could affect the biological activity of NPs. Future research may also investigate the fragility of the biocompound layer stabilizing the NPs. Laboratory processes such as centrifugation might affect this coating.

We also presume that, in the near future, computational modelling will significantly contribute to the deciphering of the complex relationships between all the interrelated variables that influence the formation of NPs. Research involving artificial neural networks for the prediction of bio-inspired NP characteristics has already emerged [242,243] and will possibly boost the scaling of NPs green synthesis to the industrial level. Referring to all the aspects discussed, we consider that the potential of green NPs is a long way from being fully explored.

## Figures and Tables

**Figure 1 molecules-27-06472-f001:**
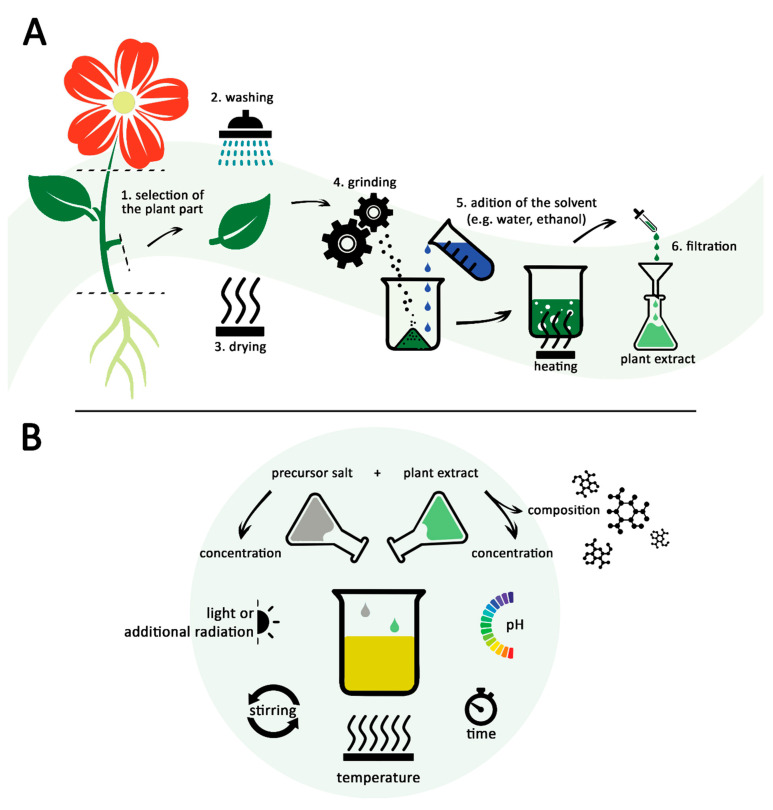
The typical approach of green synthesis of NPs: (**A**) Preparation of the plant extract; (**B**) factors affecting the reaction of green synthesis and the characteristics of the resulted NPs.

**Table 1 molecules-27-06472-t001:** The maximum absorption peak of different NPs synthesized via green chemistry.

NPs Type	Maximum Absorbance Range (nm)	Absorption Peak Observed Experimentally (nm)	References
Ag	400–450	402	[54]
		415	[55]
		432	[56]
		443	[52]
Au	530–550	534	[57]
		540	[58]
		544	[59]
		549	[60]
Pd	<300	268	[61]
		293	[62]
		<300	[63,64]
Pt	260–295	262	[65]
		263	[66]
		269	[67]
		295	[68]
Cu	550–590	535	[69]
		555	[70]
		580	[71]
		581.3	[72]
Fe_3_O_4_	360–410	365	[73]
		405	[74]
		410	[75]
TiO_2_	310–360	315	[76]
		320	[77]
		322	[78]
		355	[79]
ZnO	360–380	360	[80]
		364	[81]
		370	[82]
		375	[83]
NiO	300–350	300	[84]
		319	[85]
		328	[86]
		348	[87]
MgO	250–290	250	[88]
		260	[89]
		270	[90]
		282	[91]

**Table 2 molecules-27-06472-t002:** Different precursors for green synthesized NPs.

NPs Type	Precursors	Concentration (mM)	References
Au	chloroauric acid	0.1	[57]
Ag	silver nitrate	4	[52]
Pd	palladium chloride	0.3	[152]
	disodium tetrachloropalladate	10	[62]
	palladium acetate	2	[153]
Pt	chloroplatinic acid	1	[154]
Cu	copper sulphate	10	[155]
	copper chloride	1000	[156]
	copper nitrate	0.1	[157]
	copper acetate	100	[158]
Fe-oxides	iron nitrate	100	[131]
	iron chloride	1	[159]
	iron sulphate	100	[160]
TiO_2_	bulk titanium dioxide	5	[161]
	titanium tetraisopropoxide	100	[162]
	titanium oxysulfate	500	[163]
	titanium tetrachloride	1000	[164]
	metatitanic acid	5	[165]
	titanium butoxide	400	[166]
ZnO	zinc acetate	2–20	[129]
	zinc nitrate	1000	[167]
	zinc sulphate	1	[168]
NiO	nickel nitrate	300	[169]
	nickel acetate	100	[170]
MgO	magnesium nitrate	1170	[171]
	magnesium acetate	500	[172]
	magnesium chloride	1	[173]

**Table 3 molecules-27-06472-t003:** Studies exploring metallic/metallic oxide NP synthesis through one-molecule green approaches.

Used Biomolecule	NPs type	Used Precursor	Size (nm) and Shape	Reference
epigallocatechin-3-gallate	Au	sodium tetrachloroaurate	10.02 ± 2.5; spherical	[214]
	Ag	silver nitrate	31.67 ± 8.38; irregular	[215]
resveratrol	Au	chloroauric acid	~10; spherical	[216]
		sodium tetrachloroaurate	56.1; spherical	[34]
curcumin	Ag	silver nitrate	12.6 ± 3.8; spherical	[35]
gallic acid	Ag	silver nitrate	35–79; spherical	[217]
	Au	chloroauric acid	18–59; spherical	
			30.3 ± 3.98; spherical	[218]
tannic acid	Ag	silver nitrate	43.56 ± 4.67; spherical	[219]
Vanillin	Au	chloroauric acid	35; hexagonal, triangular, spherical	[220]
	TiO_2_	titanium tetraisopropoxide	500; spherical	[221]
Caffeine	Au	chloroauric acid	77 ± 5; spherical	[222]
cannabidiol	Ag	silver nitrate	4.82 ± 2.04; spherical	[223]
	Au	chloroauric acid	8.40 ± 5.50; spherical	
Eugenol	Ag	silver nitrate	20–30; cubic	[224]
Lycopene	TiO_2_	titanium butoxide	80–250; spherical	[225]
	Se	sodium selenite	129.3; spherical	[226]
rosmarinic acid	Au	chloroauric acid	30.46 ± 6.25; mostly spherical (also triangular, hexagonal and pentagonal)	[227]
	Ag	silver nitrate	2–5; spherical	[228]
Luteolin	TiO_2_	titanium trichloride	33.3–135; rod, prismatic, spherical, polygonal	[229]
quercetin	Ag	silver nitrate	8.4 ± 0.3; spherical	[33]
silymarin	Au	chloroauric acid	4–11; spherical	[230]
Apigenin	Se	sodium selenite	124.3; spherical	[231]
	Au	chloroauric acid	19.1 ± 10.4; spherical	[232]
β-carotene	Ag	silver nitrate	60 ± 5; triangular, polyhedral	[233]
Crocin	Au	chloroauric acid	1–10; spherical	[234]
Chitosan	Ag	silver nitrate	21; triangular, spherical	[235]
	Au	chloroauric acid	7.84 ± 2.53; spherical	[236]
Pullulan	ZnO	zinc nitrate	28.86 ± 15.46; spherical, hexagonal	[237]
Sucrose			32–40; spherical	[238]

## Data Availability

Not applicable.

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
