# Peer review of "New Green Approaches in Nanoparticles Synthesis: An Overview"

_molecules, 2022, doi:10.3390/molecules27196472_

Round 1

Reviewer 1 Report

Article review: New green approaches in nanoparticles synthesis: an overview

The review is thoroughly prepared and beneficial for nanotechnologists focusing on the biotechnological synthesis of metal and metal-containing nanoparticles.

I kindly ask the authors to edit manuscript according to the recommendations, I hope that this will contribute to the result, which will be a highly cited, quality review. After completion, I recommend for acceptance for publication.

1. I ask the authors to emphasize the fact that in the case of some compounds, mostly oxides such as ZnO or TiO2 or CuO, it is probably a process other than redox. If there is an article that deals with the mechanism of the biotechnological process of the formation of these oxides, please provide the citation and specify the formation of such nanoparticles in chapter 2. mechanism of green synthesis. It is necessary to distinguish the bioreduction of metals such as typically Ag, Au, Pd, Pt... and the biosynthesis of oxides. Of course, redox processes can occur in parallel in biomolecules, but regarding used precursor and reaction product, there is no change in the oxidation state of Zn, Ti, Cu, etc..

2. Line 14, 15 – too general…, the same in Line 53-56

3. Line 76-77: At what temperature does sintering take place? Doesn't particle growth then occur? What are other methods of separating nanoparticles from the cell? Is it always necessary? For some applications, e.g. in catalysis, palladized cells can be used. Please add citations and clarify.

4. Line 81-86: According to the current description, the in vitro and extracellular mechanism is not very well understandable, please add the examples.

5. L87: beforementioned?? Better aforementioned

6. Tab 1: Pd-examples are missing

7. Tab 1: Cu/CuO should be instead Cu, ref 44, 45 should be replaced, unfortunately, these articles are not very informative, at least due to the very poor quality data from electron microscopy.

8. L123: Electron microscopy instead of electric

9. At the end of the chapter on the mechanism, you describe the characterization methods. I would either put them in a separate chapter, or then elaborate more on the FTIR method (only functional groups!) and add chromatography (HPLC), because it is crucial for determining the mechanism, especially when working with plant extracts. Protein sequencing, when working with microorganisms, is not mentioned at all. If someone is able to determine exactly from FTIR alone which biomolecules are responsible for the biosynthesis of nanoparticles, then I consider it a masterpiece (L530-532).

10. The name of the 3. Chapter should be changed, because it is not always redox process, see comment 1)

11. L160: The term "spherical" is misleading and it looks like the authors took this information from an article that misinterprets the data. The word spherical in ref 170 is only mentioned in the abstract and not elsewhere in the text. I ask the authors to always read the entire article and critically evaluate the image attachment of the article to see if it corresponds to the discussed data. Alternatively, if, for example, the electron microscope images are of low quality (ref 44, 45) and do not support what the authors claim, another article must be cited.

12. There are publications that also describe the use of brown algae (diatoms) or algae with siliceous structures in the biosynthesis of Au and Ag nanoparticles. The advantage is the anchoring of nanometals on biosilica and the possibility of further use of the silica+nanometal combined material (diatom nanotechnology). I ask the authors to add this info to chapter 3.4.

13. In chapter 4.1. the concentrations of precursors used should also be mentioned (add a column to the table), as this is also a very important factor.

14. Table 2: FeO is not probably accurate, Fe-oxides maybe will be more correct for oxides of Fe.

15. L358: In the case of bioreduction of gold, the reaction sometimes occurs immediately after mixing, the time is certainly shorter than the stated 5 minutes... e.g. some algae sp. + HAuCl4...

16. L372: So how to prevent further particle growth, what methods are used? Centrifugation (problem with disrupting the bio-stabilization coating?) or fluidic synthesis on a chip with an oil phase, additional stabilization?? If possible, add at least general information with some review cited.

17. L379: some other more general paper should be cited here, or specify why this particular publication belongs here

18. L392: lower pH instead of higher

19. L394: “extreme points” is not technical, probably limiting values…

20. 408-413: delete “one”, improve English

21. 419-420: These results could be due to the zeta potential of biogenic NPs synthesized at different pH values. - Technically incorrect sentence

22. 437-439: This statement is too general and misleading. There are many articles that describe the biosynthesis of gold nanoparticles at room temperature.

23. 472: I would not use the term biogenic, even though it often appears in publications. This is a technology of laboratory preparation - bionanotechnology, it is not a process by which living organisms intentionally create biogenic nanoparticles for the purpose of some function, although in many cases it is a defense process.

24. Chap. 4.6. However, nanosilver is photosensitive, which means that under the influence of light, nanoparticle growth and their aggregation occur. So how does this fact go together with the lighting condition during synthesis. Can you please clarify or add information?

25. 639-641: This claim is not supported by any citation and thus seems implausible.

Author Response

Response to Reviewer #1

Article review: New green approaches in nanoparticles synthesis: an overview

The review is thoroughly prepared and beneficial for nanotechnologists focusing on the biotechnological synthesis of metal and metal-containing nanoparticles.

I kindly ask the authors to edit manuscript according to the recommendations, I hope that this will contribute to the result, which will be a highly cited, quality review. After completion, I recommend for acceptance for publication.

  1. I ask the authors to emphasize the fact that in the case of some compounds, mostly oxides such as ZnO or TiO2 or CuO, it is probably a process other than redox. If there is an article that deals with the mechanism of the biotechnological process of the formation of these oxides, please provide the citation and specify the formation of such nanoparticles in chapter 2. mechanism of green synthesis. It is necessary to distinguish the bioreduction of metals such as typically Ag, Au, Pd, Pt... and the biosynthesis of oxides. Of course, redox processes can occur in parallel in biomolecules, but regarding used precursor and reaction product, there is no change in the oxidation state of Zn, Ti, Cu, etc..

Response: Thank you very much for your remark. Explanation of the mechanism of formation of metallic oxide particles was added in chapter 2, lines 169-188.

  1. Line 14, 15 – too general…, the same in Line 53-56

Response: The sentence from lines 14-15 was rephrased. We removed the phrase: ‘Even though some progress has been done in deciphering how the characteristics of NPs are changed by modulating all the factors involved, there is still a lack of knowledge and in some cases conflicting data pose difficulties to researchers’ from lines 66-69 of the revised manuscript.

  1. Line 76-77: At what temperature does sintering take place? Doesn't particle growth then occur? What are other methods of separating nanoparticles from the cell? Is it always necessary? For some applications, e.g. in catalysis, palladized cells can be used. Please add citations and clarify.

Response: Thank you very much for your remarks. The temperature applied in sintering is below the melting point of the metal extracted, therefore we thought this process consists in the incineration of the biomass without affecting nanoparticles. However, we found sintering is used for compacting powders, while some of the particles might maintain their integrity. Therefore, in the revised manuscript we considered sintering as a method to purify metals from plants, but not nanoparticles. You can find more clarifications between lines 105-122.

  1. Line 81-86: According to the current description, the in vitro and extracellular mechanism is not very well understandable, please add the examples.

Response: We extended the description of the 2 mechanisms. You can find the additional information between lines 129-142.

  1. L87: beforementioned?? Better aforementioned

Response: Corrected as suggested.

  1. Tab 1: Pd-examples are missing

Response: Pd-examples were added in Table 1.

  1. Tab 1: Cu/CuO should be instead Cu, ref 44, 45 should be replaced, unfortunately, these articles are not very informative, at least due to the very poor quality data from electron microscopy.

Response: Correction was made and the 2 references were replaced.

  1. L123: Electron microscopy instead of electric

Response: Corrected as suggested.

  1. At the end of the chapter on the mechanism, you describe the characterization methods. I would either put them in a separate chapter, or then elaborate more on the FTIR method (only functional groups!) and add chromatography (HPLC), because it is crucial for determining the mechanism, especially when working with plant extracts. Protein sequencing, when working with microorganisms, is not mentioned at all. If someone is able to determine exactly from FTIR alone which biomolecules are responsible for the biosynthesis of nanoparticles, then I consider it a masterpiece (L530-532).

Response: The most common techniques for the characterization of green nanoparticles were separated in a new chapter (page 5). We added additional information about the FTIR method and mentioned the importance of HPLC and protein sequencing for the studies dealing with green synthesis (lines 216-233). We reformulated the phrase (lines 671-675) to correctly denote the cited results.

  1. The name of the 3. Chapter should be changed, because it is not always redox process, see comment 1)

Response: Name of the new chapter 4 was changed (page 7).

  1. L160: The term "spherical" is misleading and it looks like the authors took this information from an article that misinterprets the data. The word spherical in ref 170 is only mentioned in the abstract and not elsewhere in the text. I ask the authors to always read the entire article and critically evaluate the image attachment of the article to see if it corresponds to the discussed data. Alternatively, if, for example, the electron microscope images are of low quality (ref 44, 45) and do not support what the authors claim, another article must be cited.

Response: The misleading information from reference 70 was eliminated (line 268).

  1. There are publications that also describe the use of brown algae (diatoms) or algae with siliceous structures in the biosynthesis of Au and Ag nanoparticles. The advantage is the anchoring of nanometals on biosilica and the possibility of further use of the silica+nanometal combined material (diatom nanotechnology). I ask the authors to add this info to chapter 3.4.

Response: The suggested information was added in the chapter 4.4 of the revised manuscript (lines 319-327).

  1. In chapter 4.1. the concentrations of precursors used should also be mentioned (add a column to the table), as this is also a very important factor.

Response: Concentrations of the precursors were added in Table 2.

  1. Table 2: FeO is not probably accurate, Fe-oxides maybe will be more correct for oxides of Fe.

Response: Corrected as suggested.

  1. L358: In the case of bioreduction of gold, the reaction sometimes occurs immediately after mixing, the time is certainly shorter than the stated 5 minutes... e.g. some algae sp. + HAuCl4...

Response: We reformulated the statement (lines 489-490).

  1. L372: So how to prevent further particle growth, what methods are used? Centrifugation (problem with disrupting the bio-stabilization coating?) or fluidic synthesis on a chip with an oil phase, additional stabilization?? If possible, add at least general information with some review cited.

Response: Unfortunately, there are no reviews dealing with the prevention of particle growth and there is no information regarding the influence of centrifugation on the bio-stabilization coating. However, most studies used centrifugation in order to isolate nanoparticles and still reported the presence of functional groups on their surface. Normally, particle growth stops when the green synthesis reaction is completed and the capping process is efficient – for preventing aggregation. We suppose that precursor salt: biological agent ratio can be optimized in order to achieve a complete reaction when nanoparticles have the desired size.

  1. L379: some other more general paper should be cited here, or specify why this particular publication belongs here

Response: More general papers were cited (line 512).

  1. L392: lower pH instead of higher

Response: In order to avoid confusion, we changed ‘higher’ to ‘alkaline’ (line 524). We referred to the previously mentioned alkaline reaction mixture that, due to the activation of phytochemicals, produce smaller size nanoparticles. Some examples are given further in the manuscript.

  1. L394: “extreme points” is not technical, probably limiting values…

Response: Corrected (lines 526-527).

  1. 408-413: delete “one”, improve English

Response: Corrected.

  1. 419-420: These results could be due to the zeta potential of biogenic NPs synthesized at different pH values. - Technically incorrect sentence

Response: The sentence was reformulated to be technically correct (lines 553-554).

  1. 437-439: This statement is too general and misleading. There are many articles that describe the biosynthesis of gold nanoparticles at room temperature.

Response: We eliminated the generality from the statement (lines 573-577).

  1. 472: I would not use the term biogenic, even though it often appears in publications. This is a technology of laboratory preparation - bionanotechnology, it is not a process by which living organisms intentionally create biogenic nanoparticles for the purpose of some function, although in many cases it is a defense process.

Response: The term ‘biogenic’ was changed throughout the revised manuscript.

  1. Chap. 4.6. However, nanosilver is photosensitive, which means that under the influence of light, nanoparticle growth and their aggregation occur. So how does this fact go together with the lighting condition during synthesis. Can you please clarify or add information?

Response: We already stated ‘that only the conversion of metallic ions to NPs can be light-dependent. Adsorption of ions to phytochemicals or capping of NPs are processes that can occur even under dark conditions’ (lines 658-660). The light-induced mechanism is not fully explored but we can presume that the bio-stabilization layer could prevent nanoparticles from aggregation.

  1. 639-641: This claim is not supported by any citation and thus seems implausible.

Response: We added citations in order to support our claim (line 802).

Reviewer 2 Report

Some comments to the author's consideration

1. Authors should be consistent with the language choice sometimes they use American English in other places British English

2. The literature is full of reviews focusing on the green synthesis of nanoparticles from various biological sources. authors should clearly state the novelty of the approach and the added scientific value

3. Authors should discuss the phytochemical content of plants and their role in the formation and the mechanism in which such nanomaterials are formed. 

4. what is the added value of the green synthesis as it is very difficult to control the size of the generated materials like the chemical reduction of the metal salts therefore it will be a broad range of sizes which will be a big issue in the nanomedicine applications from drug delivery to molecular imaging

5. In figure 1 it is suggested that green synthesis could control the pH or does the author indicate by controlling the pH will influence the formation of nanomaterials

6. Line 120, the Authors suggest that UV could provide reliable information about the shape, size, and concentration. It is very difficult or near impossible to determine the shape of nanomaterials based on the UV spectrum or absorbance. agglomeration can be detected either visually if there is precipitation or by looking into the correlation graph of the DLS measurements but not from UV. This is misleading so please revise 

7.  Line 132, it is widely agreed upon and reported that the absolute value of I25I is deemed stable outside this range it will be moderate stable or unstable. also, authors should comment on what is the influence of charge on particles' stability 

8. reference 69 doesn't seem to fit with the statement stated by the authors

9. what are the differences in particles' nature and their properties between using microorganisms or plant extracts

10. in the reported nanomaterials from plants there are the wide list that has not been included like Ziziphus, banana extract and various other plants 

maybe authors should consider a table for this section

11. it has been reported that metallic salt type can lead to different forms and shapes of nanomaterials please add a paragraph on this 

12. be consistent sometimes you write the world gold or silver followed by NPs and others you write AuNPs

13. What is the future prospect of the green synthesis authors should include a paragraph at least for this 

14. what are the potential benefits for green synthesis of nanomaterials

 15. English language has to be revised for both accuracy and grammar mistakes

Author Response

Response to Reviewer #2

Some comments to the author's consideration

  1. Authors should be consistent with the language choice sometimes they use American English in other places British English

Response: Thank you very much for your remark. The manuscript was revised in accordance with American English.

  1. The literature is full of reviews focusing on the green synthesis of nanoparticles from various biological sources. authors should clearly state the novelty of the approach and the added scientific value

Response: The novelty and added scientific value were claimed in the revised manuscript, in Introduction (lines 70-83).

  1. Authors should discuss the phytochemical content of plants and their role in the formation and the mechanism in which such nanomaterials are formed.

Response: The formation mechanism was discussed more in chapter 2 (lines 169-188) of the revised manuscript. Chapter 5.7 is dedicated to biomolecules included in the phytochemical content.

  1. what is the added value of the green synthesis as it is very difficult to control the size of the generated materials like the chemical reduction of the metal salts therefore it will be a broad range of sizes which will be a big issue in the nanomedicine applications from drug delivery to molecular imaging

Response: We presented the advantages of the green synthesis in the Introduction of the revised manuscript (lines 45-58).

  1. In figure 1 it is suggested that green synthesis could control the pH or does the author indicate by controlling the pH will influence the formation of nanomaterials

Response: Obviously, the pH, as the other factors indicated in Figure 1, influences the formation of biologically synthesized nanoparticles. The description of the Figure 1B states that the displayed factors are influencing the reaction.

  1. Line 120, the Authors suggest that UV could provide reliable information about the shape, size, and concentration. It is very difficult or near impossible to determine the shape of nanomaterials based on the UV spectrum or absorbance. agglomeration can be detected either visually if there is precipitation or by looking into the correlation graph of the DLS measurements but not from UV. This is misleading so please revise

Response: Thank you for your remark. We corrected the misleading information in the chapter 3 of the revised version of the manuscript.

  1. Line 132, it is widely agreed upon and reported that the absolute value of I25I is deemed stable outside this range it will be moderate stable or unstable. also, authors should comment on what is the influence of charge on particles' stability

Response: The zeta potential value was corrected as suggested (line 238).

Thank you for your suggestion, but unfortunately there is no data regarding the influence of charge on nanoparticles’ stability in the scientific literature. If the reviewer referred to colloidal stability, this is discussed in chapter 3 of the revised manuscript (lines 235-240).

  1. reference 69 doesn't seem to fit with the statement stated by the authors

Response: The statement was revised and corrected (lines 257-259).

  1. what are the differences in particles' nature and their properties between using microorganisms or plant extracts

Response: Unfortunately, there is no information regarding the differences in particles`nature between plant- and microorganisms-synthesized nanoparticles. Theoretically, particles’ nature should not differ, but the bio-stabilization layer of nanoparticles is specific to the compounds found in plants or microorganisms. Therefore, nanoparticles’ biological properties should be different based on the biomolecules in the capping layer.

  1. in the reported nanomaterials from plants there are the wide list that has not been included like Ziziphus, banana extract and various other plants

maybe authors should consider a table for this section

Response: The aim of the chapter 4 in the revised manuscript was to highlight the remarkable diversity of microorganisms and plants involved so far in the biological synthesis process. We gave some examples of the most used plants and claimed that many others were used (line 361). We hope that now the reviewer will be pleased.

  1. it has been reported that metallic salt type can lead to different forms and shapes of nanomaterials please add a paragraph on this

Response: More information regarding the influence of the precursor were added in chapter 5.1 (lines 444-449 and lines 458-462) of the revised manuscript.

  1. be consistent sometimes you write the world gold or silver followed by NPs and others you write AuNPs

Response: This issue was corrected throughout the text.

  1. What is the future prospect of the green synthesis authors should include a paragraph at least for this

Response: Future prospects were added in the chapter 6 of the revised manuscript.

  1. what are the potential benefits for green synthesis of nanomaterials

Response: Benefits of the green synthesis were added in the Introduction (lines 46-58) of the revised manuscript.

  1. English language has to be revised for both accuracy and grammar mistakes

Response: We apologize for the English issues. The English language spellcheck has been performed throughout the manuscript. A full English spellcheck and editing will be performed by a native speaker if the manuscript is accepted for publication (English editing service offered by MDPI). 

Round 2

Reviewer 2 Report

No comments